# Patch-Based Difference-in-Level Detection with Segmented Ground Mask †

Yusuke Nonaka [1,*], Hideaki Uchiyama [1,2], Hideo Saito [1], Shoji Yachida [3] and Kota Iwamoto [3]

1   Graduate School of Science and Technology, Keio University, Yokohama 223-8522, Japan
2   Information Science, Science and Technology, Nara Institute of Science and Technology, Nara 630-0192, Japan
3   Visual Intelligence Research Lab., NEC Corporation, Kawasaki 211-8666, Japan
*   Correspondence: yuusukeole@keio.jp
†   This paper is an extended version of our paper published in International Symposium on Visual Computing under the title "Difference-in-level Detection from RGB-D Images".

**Abstract:** Difference-in-level detection in outdoor scenes has various possible applications, including walking assistance for blind people, robot walking assistance, and mapping the hazards of factory premises. It is difficult to detect all outdoor differences in level, such as RGB or RGB-D images, not only including road curbs, which are often targeted for detection in automated driving, but also differences in level on factory premises and sidewalks, because the pattern of outdoor differences in level is abundant and complex. This paper proposes a novel method for detecting differences in level from RGB-D images with segmented ground masks. First, image patches of differences in level were extracted from outdoor images to create the dataset. The change in the normal vector of the contour part on the detected plane is used to generate image patches of the difference in level, but this method strongly depends on the accuracy of planar detection, and it detects only some differences in level. Then, we created the dataset, consisting of image patches and including the extracted differences in level. The dataset is used for training a deep learning model for detecting differences in level in outdoor images without limitations. In addition, because the purpose of this paper is to detect differences in level in outdoor walking areas, regions in the image other than the target areas were excluded by the segmented ground mask. For the performance evaluation, we implemented our algorithm using a modern smartphone with a high-performance depth camera. To evaluate the effectiveness of the proposed method, the results from various inputs, such as RGB, depth, grayscale, normal, and combinations of them, were qualitatively and quantitatively evaluated, and Blender was used to generate synthetic test images for a quantitative evaluation of the difference in level. We confirm that the suggested method successfully detects various types of differences in level in outdoor images, even in complex scenes.

**Keywords:** deference-in-level detection; segmentation; supervised learning; classification; outdoor navigation; CNN network

## 1. Introduction

Every smartphone has cameras which are used for capturing what happens in the world. In such a smartphone, image processing is commonly performed for enhancing visibility [1], extending resolution [2,3], deblurring images [4], etc. Detection and segmentation of images of urban and natural scenes [5,6] are also very essential techniques for understanding environments, and have applications such as navigating vehicles and people with the support of smartphone cameras.

The detection of differences in level on the ground has various possible applications, including walking assistance for blind people, elderly people, factory workers, etc. In an autonomous driving scenario, the detection of differences in level on roads, such as curbs, has already been studied as a fundamental problem in autonomous driving tasks [7–9];

however, it relies on rich and large-sized vehicle-mounted sensors, including a camera and a three-dimensional light detection and ranging (3D-LiDAR), which are practically impossible to be mounted on walking people. In this paper, we aim to detect differences in level that exist in all kinds of situations, not only on roadways, but also on factory premises and on sidewalks. These differences in level can be extremely dangerous for blind and elderly people when walking, and pose a serious obstacle to the autonomous robots in our living spaces. Hence, the purpose of this paper is to model the characteristics of differences in level and to detect outdoor differences in level. As shown in Figure 1, we define a difference in level as a region on a walkable surface where the level of the surface changes discontinuously.

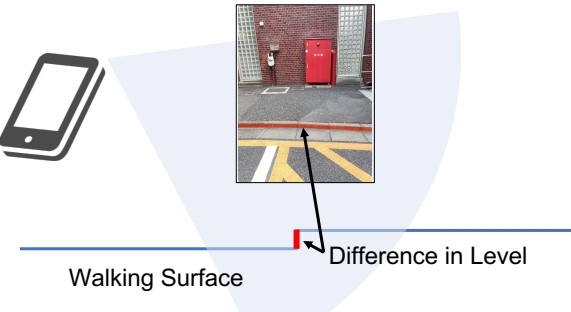

**Figure 1.** Illustration of an example of a difference in level.

As one difference-in-level detection method, Imai et al. suggested detecting walkable planar surfaces and detecting differences in level using RGB-D cameras [10]. However, they mounted an RGB-D camera at a certain level to measure the distance to the walkable planar surface, and they considered any change in height to be a difference in level. Thus, if the body posture is changed or the ground is curved, there is a high possibility of the false detection of a difference in level. Yanagihara et al. applied a convolutional neural network (CNN) and gradient-weighted class activation mapping (Grad-CAM) [11] for difference-in-level detection of a scene [12]. Since this method visualizes the basis for the CNN's decision when classifying images, with and without differences in level, and only uses image values, there are problems with detecting areas that are clearly have no differences in level and that are too wide. Consequently, the problem with conventional methods is that it is difficult to detect differences in level in outdoor scene images in which the ground is curved, or in which there are multiple types of differences in level in the images. In consideration of the many types of differences in level, especially on outdoor surfaces, detecting differences in level is still a difficult task. Most conventional methods simply use the structural features of the difference in level, such as the difference in height between two planes, though this method relies strongly on depth sensing. In a "walking" scenario, it is not possible to carry depth cameras with sufficient accuracy to detect the differences in level that hinder safe walking. Therefore, a better approach would be to acquire all the features in an RGB-D image using machine learning to determine whether there is a difference in level. With this approach, it is possible to detect various types of differences in level in several outdoor environments, as long as training data are adequately provided.

In this paper, we propose a machine learning-based method for difference-in-level detection by learning the characteristics of differences in level using a mobile depth camera. First, using 3D point clouds obtained from RGB-D images, some of the differences in level in outdoor images are extracted by detecting changes in the normal vectors in the contour of the detected walkable plane. Next, image patches of the differences in level can be automatically generated because the coordinates of the differences in level are known. Accordingly, we can create a difference-in-level dataset easily. An important note is that when we created the image patches of the normal map, we transformed the camera coordinate system into a single world coordinate system and then calculated the

normal vectors to create the dataset. Then, we trained CNNs on the dataset to detect all the differences in level, including those that could not be detected initially. In our architecture, to detect differences in level from a single image, we first divided the image into small patch images, and then we run each through a trained model. In addition, a ground mask image was generated to separate the ground from the non-ground areas in the image, and the non-target areas were removed from the image. We implemented our algorithm on an iPhone 12 Pro Max for the performance evaluation, and we conducted an ablation study with various inputs, such as RGB, depth, grayscale, normal, and combinations of them. To test whether the models were able to learn the features of differences in level, we conducted qualitative and quantitative evaluations. As a result, the best model trained with only normal vectors as inputs had almost no false positives and was able to detect successfully only differences in level in new images. This paper is an extended version of our previous paper [13], which only presents the basic framework of our method with experimental validation in limited conditions. In this paper, we have improved the accuracy by adding the segmented ground mask and enhancing the training dataset, which was evaluated by additional experiments.

To summarize, this paper will contribute the following:

- This paper proposes a method for difference-in-level detection that considers the explicit features of the differences using a machine learning technique. The proposed method accurately detects differences in level in various situations (e.g., where the ground is curved or mixed with differences in level of different sizes). In addition, as much as possible, we were able to detect differences in level using ground mask images;
- To reduce the work for manual annotation for detecting the differences in level of outdoor images, a method for automatically detecting some of the differences in level is adapted for generating datasets of image patches containing them. When creating the image patches of the normal map, the camera coordinate system of the acquired 3D point cloud was converted into a single world coordinate system, and then the normal vector was calculated. This made it easier to learn the features of the differences in level using normal vectors;
- The effectiveness of smartphones with depth cameras, which are becoming easier to use in practice, is demonstrated by experimental validation using an RGB-D camera.

## 2. Related Work

A simple method for detecting differences in level would be to identify the two planes of the ground in the image and consider the area between them as a difference in level. However, detecting planes in outdoor and complex scenes is difficult, even when using the state-of-the-art methods mentioned in Section 2.1. In Section 2.2, we also discuss two studies that have different approaches to difference-in-level detection. Then, the edge detection method, which is the basis of the proposed model, is described in Section 2.3.

### 2.1. 3D Plane Segmentation from a Single Image

Many studies have utilized planar detection in the process of the 3D reconstruction of scenes from a single RGB image. Recovering 3D planar structures from a single RGB image is also a challenging problem in 3D vision.

PlaneRCNN [14], PlaneAE [15], and PlaneTR [16] are the latest learning-based methods for planar detection from a single RGB image. PlaneRCNN and PlaneAE solve the problems of PlaneNet [17] and PlaneRecover [18], which are well-known learning-based planar detection methods, by using Mask R-CNN [19] and a deep embedding model without planar region proposals, respectively. All of these methods use context information from the CNN, as well as ignore the geometric structures in the image that are useful for 3D plane recovery. Therefore, PlaneTR uses the line segments in the image as geometric structures and detects planes using transformers [20] in conjunction with the context information in the image.

It would be desirable to be able to detect differences in level between planes using the above methods, but this is difficult to do, because these methods are adapted to indoor datasets, making it necessary to create an outdoor dataset with difference-in-level segmentation. Even if we tried to create a dataset, it would be difficult to create one that could segment all the differences in level, as there is a countless number of types and sizes, based only on RGB values. Therefore, an approach that attempts to detect differences in level directly using these planar detection methods would be difficult. By segmenting the 3D planes using the methods mentioned above, the region of the difference in level can only be detected if the area is surrounded by perfect 3D planes. However, the walking surface is not perfectly planar, but rather, curved, especially in the outdoor environment. Instead of detecting the region of the difference in level indirectly by 3D plane detection, it is better to detect it directly.

*2.2. Difference-in-Level Detection*

2.2.1. Use of RGB-D Images

As an example of difference-in-level detection for the visually impaired, Imai et al. [10] proposed a method of detecting differences in level using an accelerometer and a depth camera to detect flat surfaces. Specifically, first, the distance between the measuring device and the user's feet is measured, and it is fixed as the reference height to the current walking ground. Then, among the point clouds acquired by the measuring device, those whose normal vectors are parallel with the normal vector of the current walking plane are extracted first. The method then measures the vertical height of the extracted points and compares each point's height to the fixed reference height. If the height difference exceeds a threshold, the point is determined to be a part of the difference in level on the current walking plane. However, with this method, it is difficult to detect differences in level, unless the ground is a flat surface, which is not always the case in outdoor scenes.

Similar to difference-in-level detection, many studies have been conducted on detecting stairs as a navigational aid [21–25]. All these systems for detecting stairs focus on indoor scenes and use the geometric features of stairs. Therefore, they cannot be directly used to detect differences in level in outdoor images, which are more irregular and have smaller height differences than stairs.

Speed bump detection for autonomous driving is another research category related to our goal [26]. To take advantage of depth sensing, RGB-D cameras are often used for this purpose [27,28]. However, these methods only detect speed bumps as obstacles on roads and do not detect the location of the difference in level with a sufficient resolution.

2.2.2. Use of Convolutional Neural Network

Yanagihara et al. [12] proposed a method using CNN and Grad-CAM [11] in their study that focused on detecting small differences in level in an image. To detect level differences on roads, the method uses RGB images and CNNs. First, a CNN model that classifies road images into those with and without differences in level is trained. The basis for its decisions is visualized using Grad-CAM [11]. However, this method does not learn and detect the characteristics of the differences in level, but only classifies the presence or absence of differences in level in the image using a CNN model that is taught only from outdoor images. Therefore, even the parts that do not have obvious differences in level are judged to have differences in level.

*2.3. Edge Detection from RGB-D Images with Deep Learning*

Soria et al. proposed an advanced edge detection algorithm for RGB images [29]. Compared to edge detectors, such as in [30], a well-known method for detecting edges in low-level features, the edge detection model proposed in the paper is much more accurate. All these edge detection algorithms target only RGB images and detect all edges in the image.

Edges in an image can be divided into two types: appearance edges and occlusion edges. Appearance edges are areas in the same plane in 3D space, but in RGB images, can be readily detected as edges because of color differences. Occlusion edges, on the other hand, are edges between objects that are far apart in 3D space, but adjacent in 2D space in the image. Occlusion edge detection can be used in many applications, including object recognition, navigation, mapping, and stereoscopic vision.

An interesting study evaluating the effectiveness of deep learning for the task of occlusion edge detection is [31], wherein the occlusion edge problem is formulated and solved as a classification problem for the central pixels of an image patch with RGB-D channels. It was shown that only occlusion edges can be separated using a CNN.

## 3. Proposed Method

The aim of this paper is to learn the features of differences in level and to detect various differences in level in outdoor images. Differences in level are defined as regions on a walkable surface whose level changes discontinuously. The focus of this paper is on how the characteristics of the difference in level can be learned from RGB-D or RGB images. In addition, because this study does not yet assume real-time detection, the processing time of difference-in-level detection is not described in this paper. Section 3.1 describes the method to create a dataset for difference-in-level detection, while Section 3.2 describes the method to detect differences in level with deep learning and a ground mask.

### 3.1. Difference-In-Level Detection for Creating the Dataset

A first step in creating a dataset is to generate a 3D point cloud of the captured scene, which is done using the depth image and the camera's intrinsic parameters. Provided that the origin of the world coordinate system and that of the camera coordinate system are the same, the 3D point in the world coordinate system can be calculated for each pixel in the image.

After obtaining the 3D point cloud, the next step is to detect a plane on the walkable surface. The method for planar detection is based on [32], and it starts by estimating a plane from three points, randomly selected from the resulting 3D point cloud, and by counting the number of points within a certain distance of the plane. This process is repeated several times, and the plane with the highest number of counts is selected as the plane on the walkable surface (Figure 2I). Afterwards, the 3D points close to the surface within a threshold are detected as 3D points on the detected plane.

Then, using the camera's intrinsic parameters, the 3D points on the detected plane are projected onto the 2D image (Figure 2II) to make a binary image corresponding to the region of the detected plane in the 2D image. After morphological transformation to remove small holes from the binary image, the contour pixels of the planar area are extracted (Figure 2III).

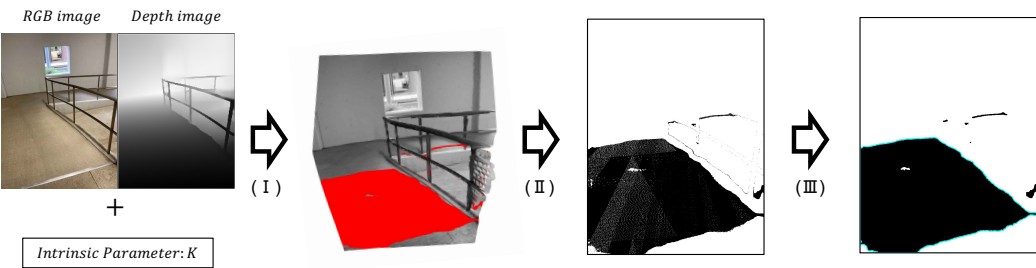

**Figure 2.** Flow of the initial difference-in-level detection method. (**I**) Create 3D point clouds from a RGB-D image. (**II**) The detected 3D plane is projected onto a 2D image using the intrinsic parameter, and then the image is binarized. (**III**) Detect the contour of the largest planar region after making the hole by morphological transformation.

Once the coordinates of the contour pixels of the detection plane are known, those of the pixels adjacent to the contour pixels can be known. Simultaneously, the coordinates, or position vectors, of each of these points in 3D space are known. By projecting the position vector onto the normal vector of the detection surface, the height difference in 3D space between the contour pixel and a pixel inside it can be computed. As shown in Figure 3, if the difference in height exceeds a certain threshold, the differences in level in the contour of the detection surface can be detected.

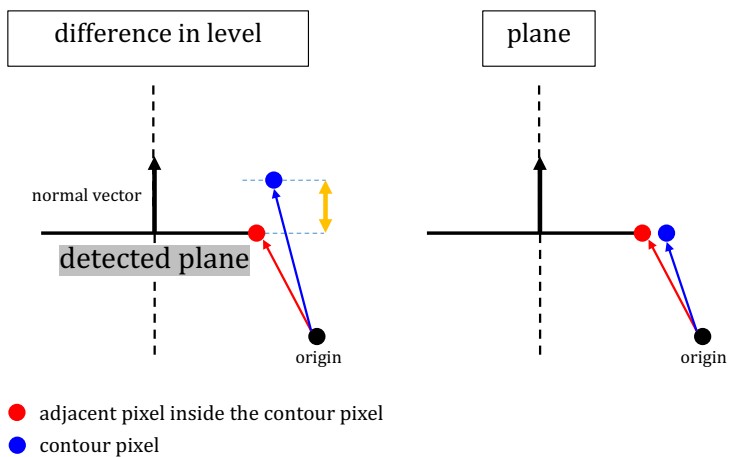

**Figure 3.** Illustration of the method to distinguish between a difference in level and a flat surface.

However, in images in which the walking surface is uneven or curved, planar detection itself is difficult, so this method can only detect some of the difference in level, not all of it.

### 3.2. Patch-Based Difference-in-Level Detection with Segmented Ground Mask

The proposed method is inspired by [31], and it determines whether the center pixels ($4 \times 4$ in this case) of a $32 \times 32$ or $64 \times 64$ image patch contain the edge of differences in level. As shown in Figure 4, according to the degree of change in the depth image, the image patches were categorized into the following three classes:

- Occlusion patch: An occlusion patch contains central pixels that have an abrupt change in value in the depth map, such as a pipe or a corner;
- Difference-in-level patch: A difference-in-level patch contains the edges of differences in level. The edge of a difference in level is defined as a boundary line where the height suddenly changes from one plane to another;
- Plane patch: A plane patch is a patch of a plane in 3D space, such as a wall or a road.

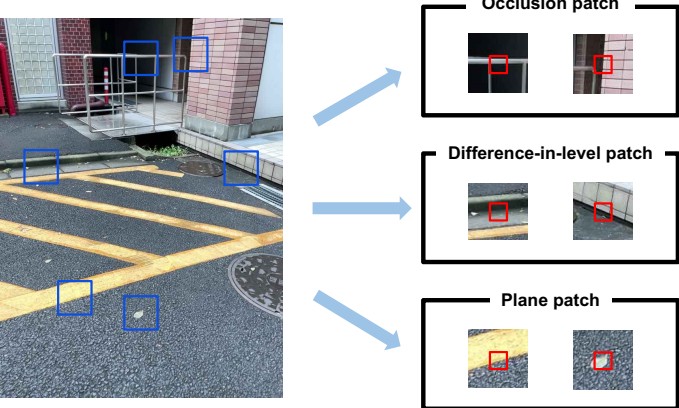

**Figure 4.** Classification of image patches.

The network has three hidden layers. The number of input channels is arbitrary, but RGB and RGB-D images are used as basic inputs in this paper. Three pairs of convolutional-pooling layers, followed by a Softmax output layer, constitute the architecture of a CNN. There are three outputs, each representing the probability that the patch is one of the three patch types described above. Thus, for example, the patch is considered to contain a difference in level if the probability of being a difference-in-level patch is higher than that of being one of the other two patch types. To compute the network loss, we used a multi-class cross-entropy error function as the loss function. In this study, the use of other models was not considered, because classification accuracy was confirmed to be sufficient with such a simple model, similar to the one used in [31].

Then, when extracting image patches from test images, mask images were used to remove all but the walking areas. For mask image generation, we trained the network in the segmentation part of [33] on the dataset in [34], which has 19 segmentation labels. However, we regarded the road and sidewalk as one label, and all the others together as one label. In doing so, a mask image was generated to separate walkable areas from the rest of the image. Then, as shown in Figure 5, areas such as walls were removed from the image to some extent. When detecting differences in level in a test image, image patches are first extracted by shifting a $32 \times 32$ or $64 \times 64$ kernel by one stride throughout the image. Each of the extracted image patches is then adapted to the training model to determine whether it is a difference-in-level patch. If either the top left or top right of the kernel contains one pixel removed by the mask, the image patch was considered to have been ignored.

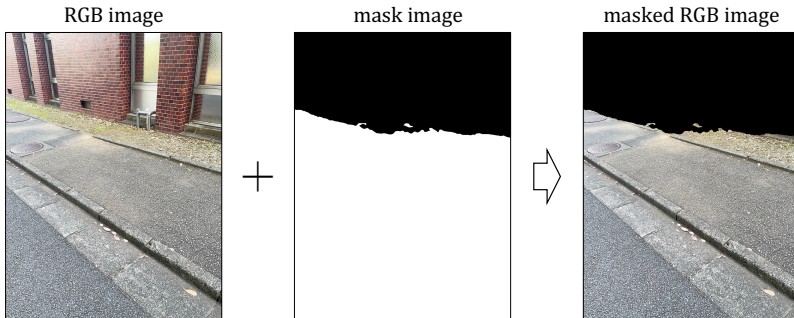

RGB image mask image masked RGB image

**Figure 5.** An example of a mask image and a masked RGB image.

## 4. Experiments

### 4.1. Dataset

The contour pixels of the detected plane were colored based on the height difference at the boundary of the detection plane, computed by the method in Section 3.1. The contour pixels of the detection plane, extracted by the method described in Section 3.1, are colored in proportion to the height difference. Specifically, areas with a height difference of 10 mm or more were colored red, areas with a difference close to zero were colored black, and areas in between were colored red in proportion to the height difference. Figure 6 shows examples of the output result described in Section 3.1. The result shows that, in the contour pixels of the detection plane, differences in level are colored red and the rest are colored black.

Image patches were extracted from outdoor images taken with an iPhone 12 Pro to create the dataset, and there are two different image patch sizes: $32 \times 32$ and $64 \times 64$. To ensure that the environment of experimental data was not too single, outdoor images were taken in 45 different scenes. Difference-in-level image patches were extracted from the outdoor images with reference to the differences in level found using the method described in Section 3.1. Occlusion patches were created by detecting edges in the depth image and creating image patches centered on the edge pixels, while plane patches were created by detecting edges in the RGB image and creating image patches centered on the edge pixels. Plane patches with no edges were collected by us. There are 206 occlusion patches, 567 difference-in-level patches, and 643 plane patches in the $32 \times 32$ image patch dataset. Further, there are 204 occlusion patches, 567 difference-in-level patches, and 643 plane

patches in the 64 × 64 image patch dataset. Examples of the dataset (patch size of 32 × 32) are shown in Figure 7. The dataset is divided into training, validation, and test datasets at a ratio of 6:2:2.

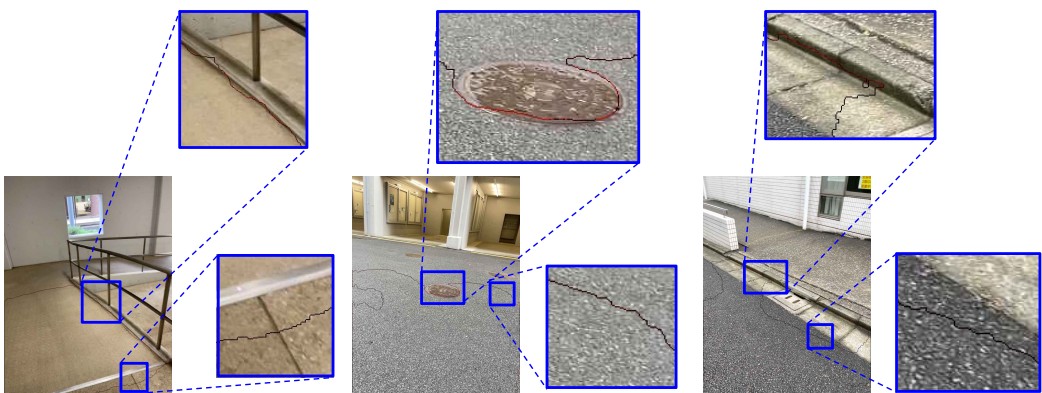

**Figure 6.** Difference-in-level areas.

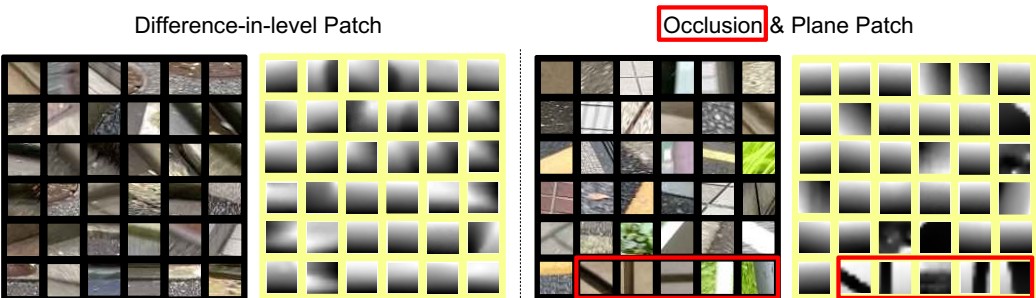

**Figure 7.** Example patch images of the dataset.

### 4.2. Network Training

The network optimizer was Adam [35] ($\beta_1 = 0.9, \beta_2 = 0.999$) with a mini-batch size of 100. The biases of the convolutional layers were all initialized to zero, while the weights were initialized with zero-mean Gaussian distributions with standard deviations of 0.0001 for the first and 0.01 for the second and third convolutional layers. For the output layer, the weight standard deviation was initialized with 0.3. The number of stride, kernel size, and padding for convolution and pooling are listed in Figure 8. The learning rate was 0.01 and the weight decay was set to 0.001. The value of the epoch was 100, and dropout was not applied when training the model. The experiments were implemented in PyTorch (v1.10.1, with Python 3.7.10) and run on an Nvidia GeForce GTX 1080 using CUDA 10.1. The settings for the network were the same as in [13].

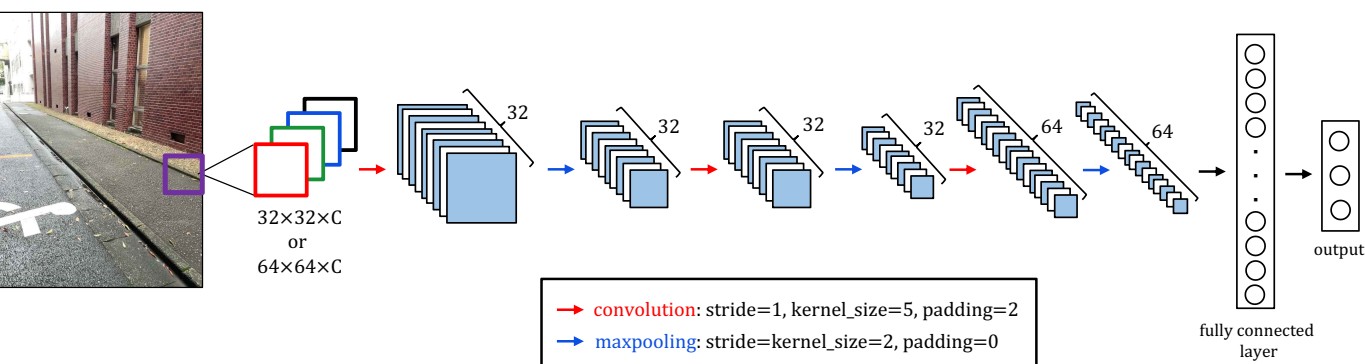

**Figure 8.** Convolutional neural network architecture for difference-in-level detection.

### 4.3. Difference-in-Level Detection on Various Inputs

There is no restriction on the number of input channels, so the network can take various input types. Different input forms can be created from RGB-D images. For example, an RGB image has three channels, but when converted to a grayscale image, it has a single channel. In addition, a depth image has one channel, but if converted to a normal map, it has three channels. To generate a normal map, a point cloud within 10 cm of the target point was extracted, and a plane in this point cloud was estimated in the same way as described in Section 3.1. The normal vector of the plane was determined to be that of the target point.

In addition, two points should be noted in the calculation of normal vectors. First, because the camera coordinate system differs if the camera is in a different location, the normal value of the walking plane changes for each camera location if the calculation is performed using the 3D coordinate without modification. Therefore, the camera coordinate system was converted to a world coordinate system with the direction of gravity parallel to the y-axis, and then the normal map was calculated. In addition, the second thing to note is that normal vectors that do not point towards the camera should be flipped to the side of the camera. Figure 9 shows examples of a normal map of a scene taken by the iPhone 12 Pro Max. Thus, as shown in Table 1, the network can receive various combinations of input data.

RGB images          Normal maps

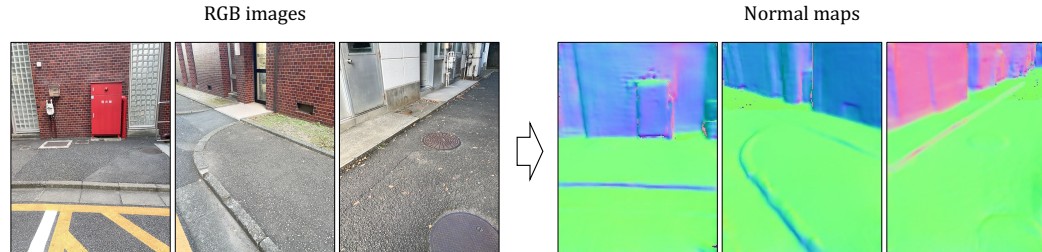

**Figure 9.** Examples of normal maps.

**Table 1.** Input variation.

|  | Channel | #1 | #2 | #3 | #4 | #5 | #6 | #7 | #8 | #9 | #10 |
|---|---|---|---|---|---|---|---|---|---|---|---|
| RGB | 3 | ✓ | ✓ |  |  |  | ✓ | ✓ |  |  |  |
| Gray Scale | 1 |  |  | ✓ | ✓ |  |  |  | ✓ | ✓ |  |
| Depth | 1 |  | ✓ |  | ✓ |  | ✓ |  |  | ✓ |  |
| Normal Vector | 3 |  |  | ✓ | ✓ | ✓ |  |  | ✓ | ✓ | ✓ |
| Channel Sum | — | 3 | 4 | 4 | 5 | 3 | 3 | 4 | 4 | 5 | 3 |
| Patch Size | — |  |  | 32 |  |  |  |  | 64 |  |  |

#### 4.3.1. Quantitative Evaluation

Table 2 shows the macro-averaged Precision, Recall, and F1-Score for the output results of the three-class classification on the test dataset. The overall result shows that models using normal vectors as inputs are more accurate than others. Comparing the $32 \times 32$ and $64 \times 64$ models overall, we can see that #3, #4, and #5 have almost the same F1-Scores, but in comparing #8, #9, and #10, only #10 is 0.1% more accurate than #8 and #9. Since there are few studies of difference-in-level detection, and we use our own definition of a difference in level, it is difficult to compare the accuracy of our method with comparative methods.

#### 4.3.2. Qualitative Evaluation

For a qualitative evaluation, we visualized differences in level. The images to be visualized were divided into small image patches, and the network took each patch. As shown in Figure 10, when the probability of being a difference-in-level patch exceeded 50%,

it was colored according to this probability. Only the top-left pixel of the center square (2 × 2) of an image patch was colored.

**Table 2.** Quantitative results where the macro-averaged Precision, Recall, and F1-Score for a test set created by dividing the dataset are shown. The numbers with a # sign correspond to the numbers in Table 1.

|                            | #1    | #2    | #3    | #4    | #5    | #6    | #7    | #8    | #9    | #10   |
|----------------------------|-------|-------|-------|-------|-------|-------|-------|-------|-------|-------|
| macro-averaged Precision   | 0.149 | 0.871 | 0.969 | 0.957 | 0.964 | 0.151 | 0.883 | 0.973 | 0.976 | 0.984 |
| macro-averaged Recall       | 0.333 | 0.835 | 0.955 | 0.957 | 0.955 | 0.333 | 0.796 | 0.968 | 0.970 | 0.981 |
| macro-averaged F1-Score     | 0.206 | 0.850 | 0.961 | 0.956 | 0.959 | 0.208 | 0.821 | 0.970 | 0.973 | 0.983 |

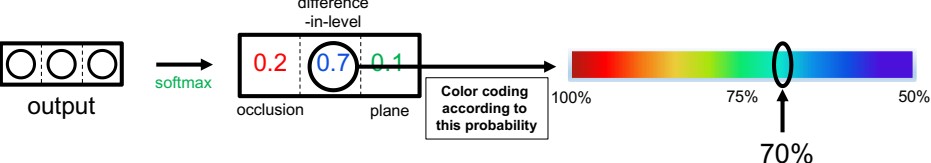

**Figure 10.** Illustration of how to visualize difference-in-level pixels.

Figures 11 and 12 show the qualitative results of the method described in Section 3.2, where some patches in the images from the top to the third are included in the training dataset, while patches in the bottom image are not.

First, #1 and #2 are compared. By adding depth values to the input, it was estimated that the features of differences in level could be learned by changes in depth values. Looking at Figure 11, it can be inferred that the result meets the expectation to some extent. We can see that the addition of the depth value to the input greatly reduces the portion of the image that is reacting to edge lines on the RGB image and misdetecting them as differences in level. However, some areas are still reacting to edges on the RGB image. Therefore, even if depth values are used as inputs, they cannot be fully utilized for difference-in-level detection.

Next, #2 and #3 are compared. Since the #3 model was trained with grayscale images and normal vectors as inputs, there is a lesser response to the color changes than with #2. The difference is best seen in the bottom image in Figure 11. Thus, it is assumed that it is better to convert the RGB or depth images to grayscale images or normal maps to learn the characteristics of the differences in level.

In #4, depth values are added to #3 with respect to the network input. By adding depth values, it is expected that the #4 model will better learn the features of the differences in level than #3. However, looking at the results of #3 and #4, it can be seen that #4 judges the flat part as a difference in level more than #3. This indicates that depth values are not necessarily required for learning differences in level. Finally, the #5 model only uses normal vectors as inputs. Overall, false positives for walls are greatly reduced for #5 compared to #3, which indicates that #5 is better at learning the features of differences in level.

The above discussion is also true for the results of the experiment using a 64 × 64 patch. However, a comparison of Figures 11 and 12 shows that Figure 12 responds more strongly to differences in level and also shows a decrease in false positives. It can be inferred that a 64 × 64 patch has a wider range of image features and is more likely to learn the features of differences in level. However, the detection area also increases as the size of the image patch increases.

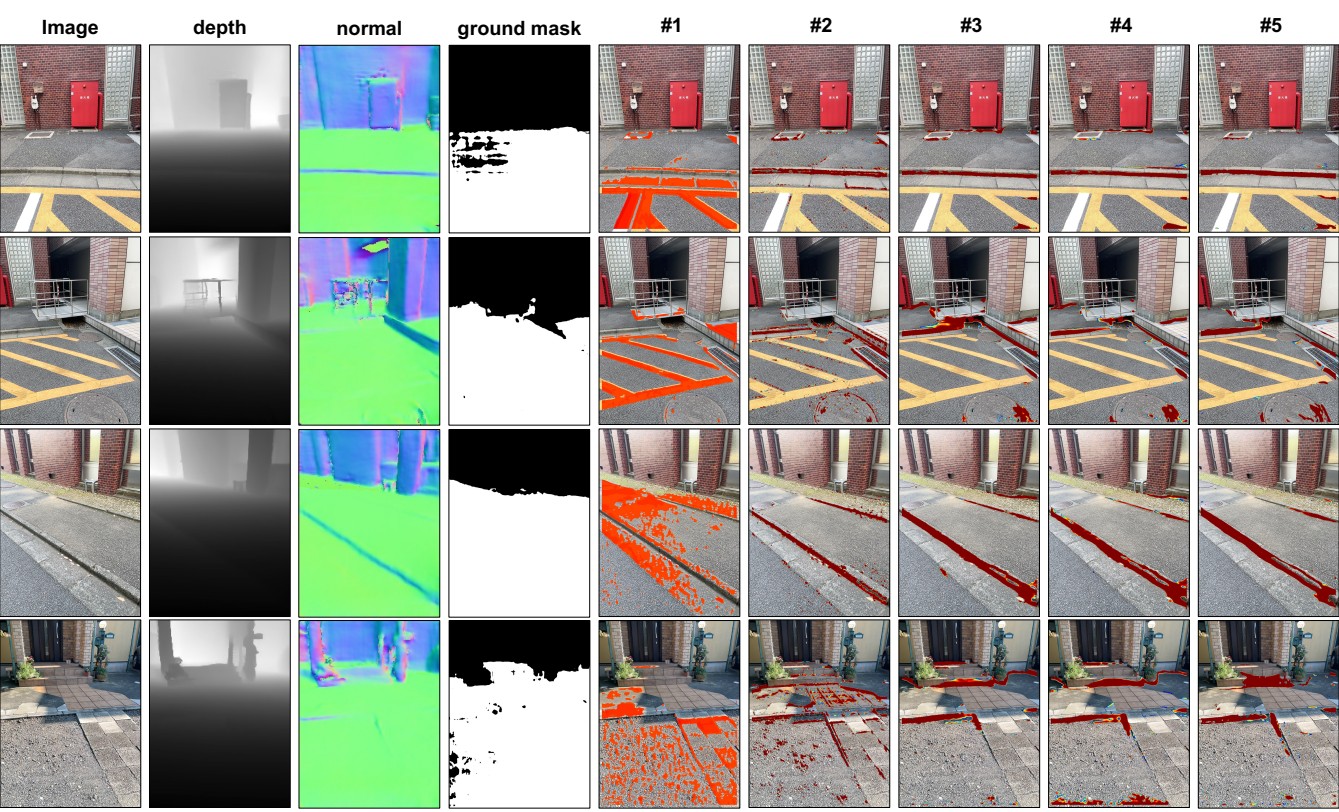

**Figure 11.** Results with a patch size of $32 \times 32$ as input. The numbers correspond to the numbers in Table 1.

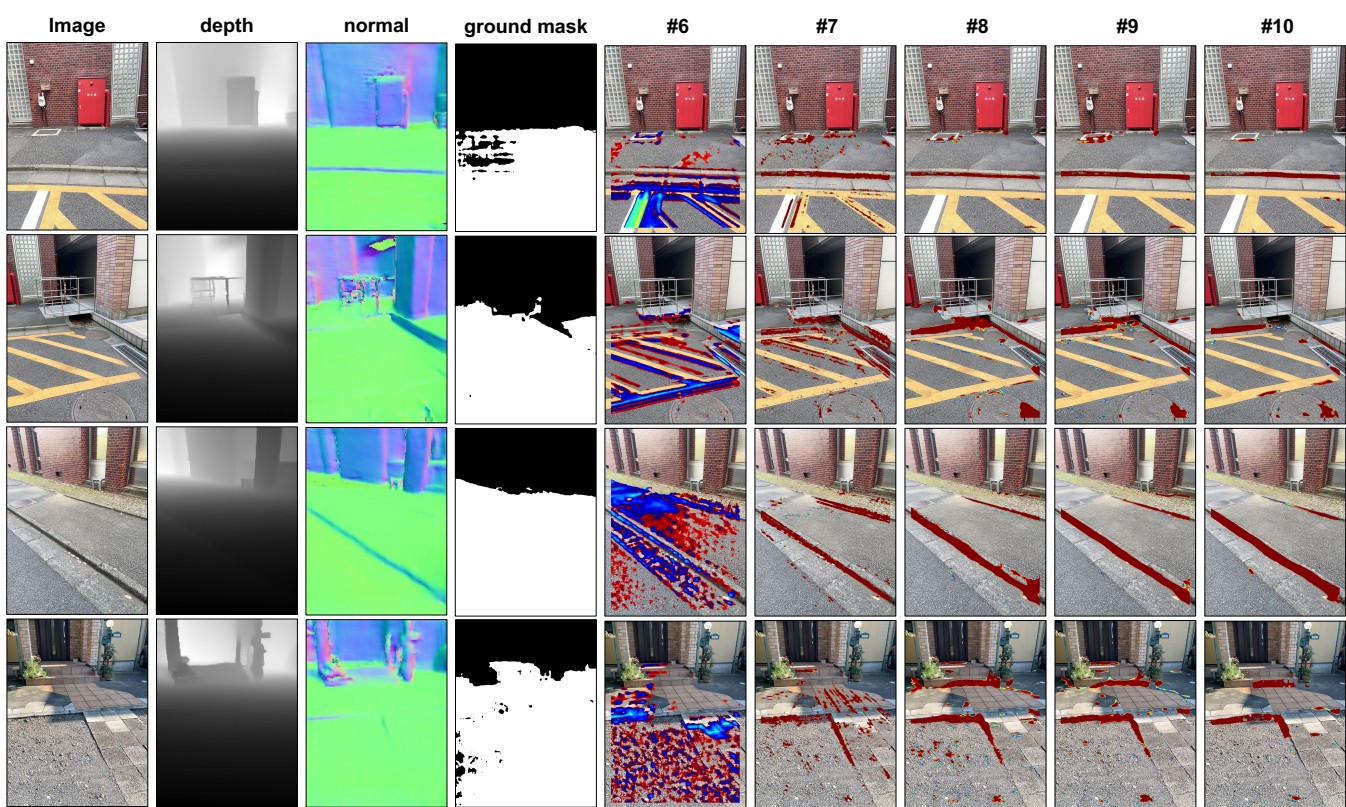

**Figure 12.** Results with a patch size of $64 \times 64$ as input. The numbers correspond to the numbers in Table 1.

Finally, we focus on the results of #10, which had the highest accuracy in Table 2. Comparing the result for the new image at the bottom of Figure 13 with #5, we see that the detection of the wall has disappeared and only the difference in level is detected. This result indicates that the model in #10 is learning and detecting the features of the difference in level.

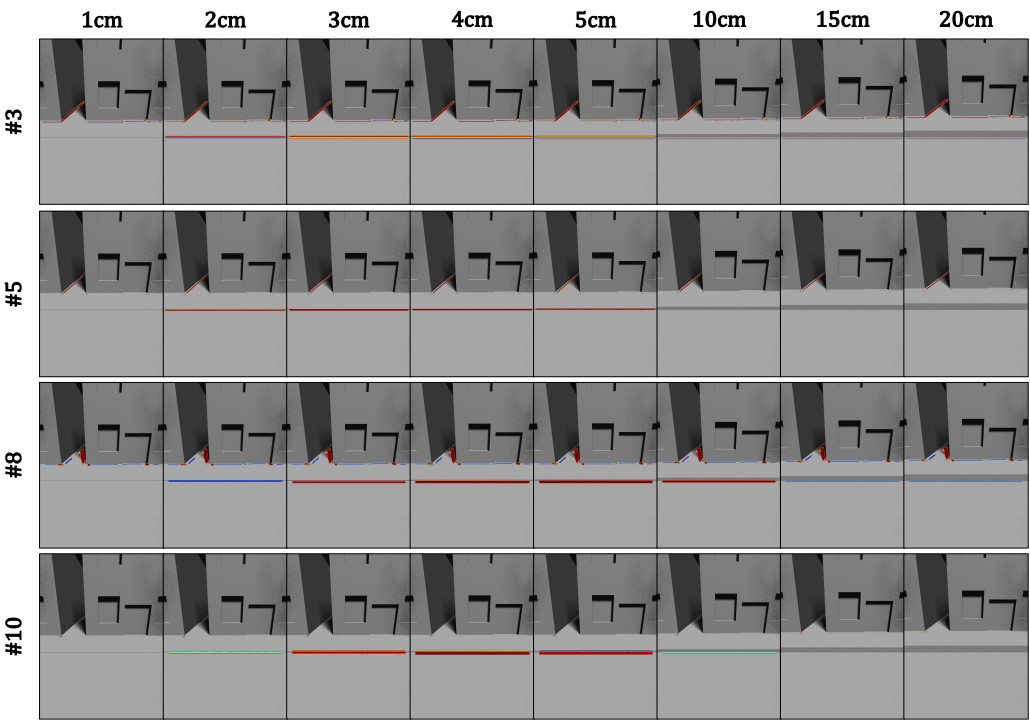

**Figure 13.** Illustration of how high a difference in level can be detected.

### 4.4. Difference-in-Level Detection for Synthetic Test Images

Then, we generated synthetic test images using Blender to see how high of a difference in level the proposed model could detect. The distance from the camera to the difference in level was about 3 m, the camera's position from the ground was about 1.37 m, the elevation angle from the ground was 71.6 degrees, and the focal length was 23.76 mm. The image size was set to 720 × 960, and the Mai city model [36] was used as the 3D model. The model to be compared here does not use RGB images as inputs, and in this experiment, we did not use a mask image of the test image due to the poor generalization across domains.

The five pictures in Figure 14 were used for the evaluation, and the result of the quantitative evaluation is in Table 3. The strength of the response was ignored, and the training model classifies into two categories: areas judged to have differences in level with a probability of more than 50% and other areas. In addition, because it is sufficient to be able to distinguish between difference-in-level patches and the rest in this study, we considered this classification to be a two-class problem and used the following three indicators to evaluate Table 3:

- Precision: The percentage of patches that are actually difference-in-level patches out of those that are predicted;
- Recall: The percentage of patches that are predicted to be difference-in-level patches out of those that are actually difference-in-level patches;
- F1-Score: The harmonic mean of Precision and Recall.

**Table 3.** Quantitative results of the Precision, Recall, and F1-Score for the 5 test images. The numbers with a # correspond to the numbers in Table 1.

|           | #3    | #4    | #5    | #8    | #9  | #10   |
|-----------|-------|-------|-------|-------|-----|-------|
| Precision | 0.331 | 0.599 | 0.512 | 0.196 | 0.0 | 0.327 |
| Recall    | 0.780 | 0.486 | 0.684 | 0.470 | 0.0 | 0.693 |
| F1-Score  | 0.456 | 0.534 | 0.560 | 0.274 | 0.0 | 0.432 |

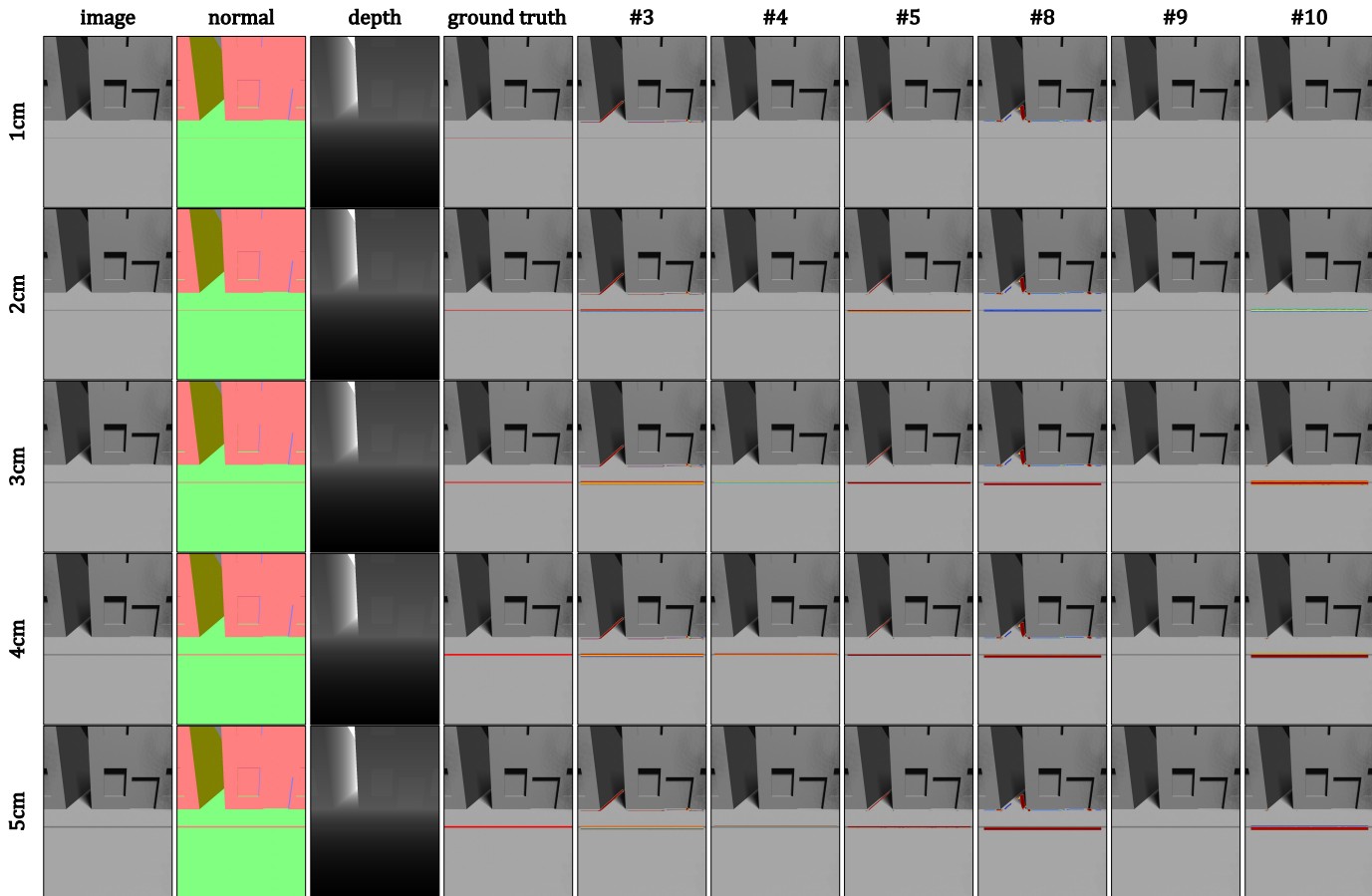

**Figure 14.** Results for synthetic test images.

Therefore, the problem setting is different from Table 2, and a model trained on a $64 \times 64$ image patch dataset detected a wider area than the ground-truth area, which reduced the accuracy of #8 and #10 in this evaluation.

Qualitative results are shown in Figure 14, which shows that the models trained with only normal vectors in #10 and #5 have relatively few false positives; #4, which uses depth values, detects differences in level from 2 cm to 4 cm, although the response is not very strong; and #9, which also uses depth values, does not detect them at all. The results of models #3 and #8, which use the normal and grayscale images as inputs, show that the difference in level can be detected to some extent, but other false detections are significant. Comparing #5 and #10 shows that #10 has little detection of the wall, similar to the results in Section 4.3. Overall, the models trained on $64 \times 64$ image patches are more responsive to detection of differences in level.

Figure 13 shows whether detection is possible with respect to differences in level that are 5 cm or greater for models #3, #5, #8, and #10. It can be seen that #8 responds slightly at heights of 15 cm and 20 cm, but the other models do not respond. Further, #10 has almost no response to walls and can detect only the difference in level, suggesting that it is learning and detecting the characteristics of the difference in level.

## 5. Conclusions

In this paper, we proposed an unprecedented difference-in-level detection method using a mobile depth camera to learn the features of differences in level. An ablation study with different input models showed that models trained with normal vectors as inputs improved the accuracy of difference-in-level detection. The best model, trained with a $64 \times 64$ image patch with normal vectors as inputs, learned the features of differences in level and was able to detect differences in level up to a certain height. However, color changes or edges in RGB images are also required to detect differences in level that are smaller than the limit of the detectable height. In the future, our goal is to create a more general and accurate model for difference-in-level detection that can detect any outdoor differences in level. The findings concerning difference-in-level detection obtained in this paper will contribute to the development of difference-in-level detection research.

**Author Contributions:** Conceptualization, Y.N., H.S., H.U., S.Y. and K.I.; methodology, Y.N., H.S. and H.U.; software, Y.N. and H.U.; validation, Y.N., H.S. and H.U.; formal analysis, Y.N., H.S. and H.U.; investigation, Y.N., H.S. and H.U.; resources, Y.N., H.S., H.U., S.Y. and K.I.; data curation, Y.N., H.S. and H.U.; writing—original draft preparation, Y.N.; writing—review and editing, Y.N., H.S. and H.U.; visualization, Y.N.; supervision, H.S.; project administration, H.S.; funding acquisition, H.S. All authors have read and agreed to the published version of the manuscript.

**Funding:** This research received no external funding.

**Institutional Review Board Statement:** Not applicable

**Informed Consent Statement:** Not applicable

**Data Availability Statement:** The data reported in this study are not available because they are private data from Keio University.

**Conflicts of Interest:** The authors declare no conflict of interest.

## Abbreviations

The following abbreviations are used in this manuscript:

| | |
|---|---|
| CNN | Convolutional neural network |
| Grad-CAM | Gradient-weighted Class Activation Mapping |

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
