# Peer review of "Patch-Based Difference-in-Level Detection with Segmented Ground Mask"

_electronics, doi:10.3390/electronics12040806_

Round 1

Reviewer 1 Report

This paper presents a novel method for detecting differences in level from RGB-D images with segmented ground masks. Experimental results show the good performance of the proposed method. However, some weaknesses should be addressed, especially the introduction and experiment.

1) In INTRODUCTION part, the introduction of background is too simple, there are too few references in this article. Many of the latest work has not been introduced, which is not enough to fully explain the significance of this study. In particular, the application of image processing technology in other fields should also be introduced. Therefore, the authors are suggested to add some literatures. e. g.,

[1] Super-Resolution Mapping Based on Spatial-Spectral Correlation for Spectral Imagery [J]. IEEE Transactions on Geoscience and Remote Sensing, 2021, 59(3): 2256-2268.

[2] Heterogeneous Knowledge Distillation for Simultaneous Infrared-Visible Image Fusion and Super-Resolution. IEEE Transactions on Instrumentation and Measurement, 2022, 71, 5004015

2) The Section 3 of the proposed method is too simple. What is the basis for selecting each module? In particular, some important parameters such as convolution layers, parameters and schematic diagram are not given in detail. It is suggested that the authors improve this part.

3) The experimental part needs to be improved, and the comparative methods are too few to reflect the progressiveness of this method. In addition, the environment of experimental data is too single, so it is suggested to add different scenario data validation methods.

4) In addition, there are some grammatical errors in the article, which need further careful proofreading.

Author Response

Thanks for the many helpful suggestions!

The following is a description of the revisions we have made to each of the comments.

1) In INTRODUCTION part, a paragraph on the application of image processing technology was added to the first paragraph.

2)
In this study, the use of other models was not considered, because classification accuracy was confirmed to be sufficient with such a simple model similar to the one used in the reference*. 
S. Sarkarand, V. Venugopalan, K. Reddy, J. Ryde, N. Jaitly, and M. Giering, “Deep learning for automated occlusion edge detection in RGB-D frames,” Journal of Signal Processing Systems, vol. 88, Aug. 2017.

The number of stride, kernel size, and padding for convolution and pooling are listed in Figure 6.

The above two sentences were added to lines 239-241 and 282-283 of the revised paper, respectively.

3) Because there are few studies of difference-in-level detection, and we use our own definition of a difference in level, it is difficult to compare the accuracy of our method with comparative methods.

To ensure that the environment of experimental data was not too single, outdoor images were taken in 45 different scenes.

The above two sentences were added to lines 313-316 and 266-268 of the revised paper, respectively.

4) To correct grammatical errors in the article, we used an editing service.

Reviewer 2 Report

Review

In this paper, the authors proposed a novel method for detecting differences in level from RGB-D images with segmented ground masks using the normal vector of the contour of the detection plane and CNN network. 

The article is interesting; however, the article requires a few explanations and additions.

1. What is the level detection rate for various combinations of input information #1 to #10 from Table 2?

2. What is the impact of increasing the accuracy of level detection on processing time? Can the method be used in real time?

3. The results could be compared with other methods or results in the literature.

4. In chapter 4.3.1. On line 282 the authors write "Table 2 shows the quantitative results according to the evaluation index described in subsection 4.3." The evaluation index is not explicitly described in Section 4.3.

5. Why are the results in Tables 2 and 3 so different?

6. CNN network can be added to Keywords.

Author Response

Thanks for the many helpful suggestions!

The following is a description of the revisions we have made to each of the comments.

1. Table 2 shows the macro-average Precision, Recall, and F1-Score values for the output results of the three-class classification for the test dataset when the dataset is divided into training, validation, and test datasets.

「Table 2 shows the macro-averaged Precision, Recall, and F1-Score for the output results of the three-class classification on the test dataset.」

The above sentence was added to lines 309-310 of the revised paper.

2. This study was not intended for real-time use. Therefore, no attempt was made to speed up the detection process.

「In addition, because this study does not yet assume real-time detection, processing time of difference-in-level detection is not described in this paper.」

The above sentence was added to lines 190-192 of the revised paper.

3. Because there are few studies of difference-in-level detection, and we use our own definition of a difference in level, it is difficult to compare the accuracy of our method with comparative methods.

The above sentence was added to lines 313-316 of the revised paper.

4. This was corrected.

5. In the synthetic image, the difference in level can be segmented as the ground-truth. Table 3 shows the Precision, Recall, and F1-Score of the two-class classification models. Therefore, the problem setting is different from Table 2, and a model trained on a 64×64 image patch dataset detected a wider area than the ground-truth area, which reduced the accuracy of #8 and #10 in this evaluation.

「Therefore, the problem setting is different from Table 2, and a model trained on a 64×64 image patch dataset detected a wider area than the ground-truth area, which reduced the accuracy of #8 and #10 in this evaluation.」

The above sentence was added to lines 377-379 of the revised paper.

6. This keyword was added.

Round 2

Reviewer 1 Report

Thanks for the authours' reply. I have no other questions.